# Synthesis and Characterization of Low-Cost Cresol-Based Benzoxazine Resins as Potential Binders in Abrasive Composites

**DOI:** 10.3390/ma13132995

**Published:** 2020-07-05

**Authors:** Artur Jamrozik, Mateusz Barczewski, Grzegorz Framski, Daniel Baranowski, Paulina Jakubowska, Łukasz Klapiszewski, Teofil Jesionowski, Adam Voelkel, Beata Strzemiecka

**Affiliations:** 1Institute of Chemical Technology and Engineering, Faculty of Chemical Technology, Poznan University of Technology, Berdychowo 4, PL-60965 Poznan, Poland; paulina.jakubowska@put.poznan.pl (P.J.); lukasz.klapiszewski@put.poznan.pl (Ł.K.); teofil.jesionowski@put.poznan.pl (T.J.); adam.voelkel@put.poznan.pl (A.V.); 2Institute of Materials Technology, Faculty of Mechanical Engineering, Poznan University of Technology, Piotrowo 3, PL-61138 Poznan, Poland; mateusz.barczewski@put.poznan.pl; 3Institute of Bioorganic Chemistry, Polish Academy of Sciences, Noskowskiego 12/14, PL-61704 Poznan, Poland; grzegorz.framski@ibch.poznan.pl (G.F.); daniel.baranowski@ibch.poznan.pl (D.B.)

**Keywords:** benzoxazine resins, phenolic resins, abrasive composites, bonding, grinding

## Abstract

A series of cresol-based benzoxazines were synthesized for potential application as a polymer matrix in abrasive composites. The chemical structures of the obtained benzoxazine resins were investigated in detail using Fourier transform infrared spectroscopy (FTIR) and hydrogen-1 as well as carbon-13 nuclear magnetic resonance spectroscopy (^1^H NMR, ^13^C NMR) with an additional analysis using two-dimensional NMR techniques (2D NMR ^1^H-^1^H COSY, ^1^H-^13^C gHSQC and gHMBC). Structural analysis confirmed the presence of vibrations of -O-C-N- at ~950 cm^−1^ wavenumber, characteristic for an oxazine ring. The thermal properties of benzoxazine monomers were examined using differential scanning calorimetry (DSC) analysis. The polymerization enthalpy varied from 143.2 J/g to 287.8 J/g. Thermal stability of cresol-based benzoxazines was determined using thermogravimetry (TGA) analysis with additional analysis of the amount of volatile organic compounds (VOC) emitted from the synthesized benzoxazines during their crosslinking by static headspace coupled with gas chromatography technique (HS-GC). The amount of residual mass significantly differed between all synthesized polybenzoxazines in the range from 8.4% to 21.2%. The total VOC emission for benzoxazines decreased by 46–77% in reference to a conventional phenolic binder. The efficiency of abrasive composites with the benzoxazine matrix was evaluated based on abrasion tests. Performed analyses confirmed successful synthesis and proper chemical structure of cresol-based benzoxazines. All the experiments indicated that benzoxazines based on different cresol isomers significantly differ from each other. Good thermal performance and stability of the abrasive composites with the polybenzoxazine matrix and significantly lower VOC emission allow us to state that benzoxazines can be a promising and valuable alternative to the phenolics and a new path for the development of modern, eco-friendly abrasives.

## 1. Introduction

Surface quality is one of the most important characteristics of products which consist of metal, stone, wood and polymers. Almost every commercial product undergoes finishing treatment at the last stage of production, which gives the appropriate dimensions, shape or texture of the surface [1]. One of the most popular methods used for this purpose is abrasive machining, the effect of which is significantly influenced by the type of abrasive tool used [2]. Proper construction and composition of the tool can guarantee the highest quality of the surface layer [3].

The bonded abrasive tools used for these purposes are usually based on a ceramic or resin binder [4]. Resinoid abrasive composites are the largest group among universal and specialized tools due to their affordable price and good performance [5]. The key component of bonded abrasive tools, apart from abrasive grains, is an organic binder which acts as a matrix. Typically, phenol-formaldehyde binders are used, but epoxy and rubber are also common [6]. Phenolics owe their popularity to a very favorable price ratio to the offered mechanical properties and good resistance to high temperature and process coolants. Despite the fact that phenol-formaldehyde resins are among the most reliable organic resin systems on the market, it should be pointed out that they have some drawbacks. The gradual release of phenol, formaldehyde and other VOCs at almost every stage of manufacturing of abrasive tools, from mixing of raw materials to hardening in furnaces, is a well-known fact [7,8,9,10]. This phenomenon also concerns many other materials containing phenolics, e.g., printed circuit boards and wood-based panels [11,12]. Taking into account the above-mentioned examples, it is important to look for any alternative that could be ecologically and economically beneficial. However, it is very difficult to define the entry level values of the binder properties because tools can differ from each other in terms of shape, type and size of the abrasive grain, as well as the use of additional fillers that can significantly change the properties of the tool. Additionally, the lack of scientific reports concerning the relationship between the matrix properties and the technical parameters of the grinding tools makes it almost impossible. For example, the adhesion between polymer matrix and the abrasive grain is a property of the tool that is very difficult to determine. It is highly a complex phenomenon because if the adhesion is too weak, the grain would fall out from the tool before it loses its sharpness, which results in a shorter life cycle. In turn, if the adhesion is too strong, it will keep the grain in the mass of the tool despite the loss of sharpness, which results in lowering the grinding efficiency and raising the temperature of the operation. All of the above-mentioned circumstances emphasize the importance of further investigation of new organic binders that were not previously utilized in such applications. The resin system has recently gained popularity and benzoxazines broaden its range of applications [13,14,15,16,17]. Benzoxazines are a type of thermoset polymers which exhibit basic properties that meet the general requirements imposed by the abrasive industry.

Benzoxazines can be an extremely interesting alternative to conventional organic resin systems. They represent a relatively new group of thermoset resins with some unique properties which can be very advantageous in abrasive technology [18,19]. Almost no polymerization shrinkage, low moisture absorption and high thermal stability are a set of desirable properties in such demanding applications [20,21,22,23,24]. The superior thermal and mechanical properties of benzoxazine resins have already been confirmed. Ortho-maleimide-substituted benzoxazines exhibit good processability and very high glass transition temperature with great inflammability [25]. Polybenzoxazines from resveratrol-based tri-functional monomers exhibit a similarly high performance characteristic. Such polymer systems exhibit low polymerization temperature, very high thermal stability and exceptionally high char yield [26]. Polybenzoxazine matrix is also a key component in an increasing number of novel high-performance composites. Composites with bisphenol-A-aniline benzoxazine matrix reinforced with silane-modified PIPD/basalt hybrid fibers resulted in a significant improvement of mechanical properties with good interfacial adhesion [27]. There are also reports regarding high thermally and mechanically stable hot-pressed bamboo/glass-reinforced polybenzoxazine hybrid composites which maintain good mechanical performance at elevated temperature [28]. The fact that application of carbon fiber as a reinforcing material for polybenzoxazine matrix using proper manufacturing process allows one to obtain modern, extremely durable composites that can be successfully implemented in highly demanding aerospace applications is also worth noticing [29].

Examples of benzoxazines with their favorable properties mentioned aforehand, such as high thermal and mechanical stability as well as good adhesion, may suggest that the polybenzoxazine matrix can be introduced in many other advanced materials [30,31,32]. The sector of abrasive composites is one of the fields which may benefit from the advantageous features of benzoxazines. In such a specific application, high thermal stability, good interfacial adhesion and inflammability are key factors used to categorize the polymer as a binder in an abrasive tool. The aim of the presented study is the synthesis and characterization of low-cost cresol-based benzoxazine resin matrix in abrasive composites as an alternative to commonly used phenolics and evaluation of potential benefits associated with their use. It is worth mentioning that benzoxazines have not been characterized for use in abrasive tools in the past.

## 2. Materials and Methods

### 2.1. Synthesis of Cresol-Based Benzoxazines

Syntheses of six cresol-based benzoxazines were performed according to a two-step Mannich reaction in toluene (Avantor Performance Materials Poland S.A., Gliwice, Poland). In the first stage, formaldehyde (34 mass % solution—Merck KGaA, Darmstadt, Germany) reacted with an amine (aniline, *p*-toluidine—Merck KGaA, Darmstadt, Germany) at reduced temperature to form N,N-bis(hydroxymethyl) amine, which, in the further step, forms benzoxazine after the addition of cresol isomer (Merck KGaA, Darmstadt, Germany). Molar ratio of the amine to formaldehyde to cresol was equal to 1:2:1, respectively. Reagents used for individual syntheses are listed in Table 1 and the schematic course of syntheses is presented in Figure 1.

The detailed description of the syntheses with use of aniline as an amine is as follows: 16.7 mL of aniline, 27.2 mL of formaldehyde solution and 10.0 mL of toluene were placed in the three neck reaction flask. The mixture was stirred at 500 rpm for 30 min at reduced temperature ca. 1–5 °C in an ice bath. After that time, 19.2 g of cresol isomer dissolved in 20 mL of toluene was added and the mixture was stirred for approx. 20 min at a reduced temperature. Afterwards, the mixture was gradually heated until it reached the temperature of approx. 84 °C, at which the water-toluene azeotrope evaporates. Using the Dean-Stark apparatus, toluene was returned to the reaction while water was removed from the mixture. The reaction was carried out for approx. 3 h until water was fully evaporated and the temperature of the mixture raised to approx. 110 °C. Afterwards the reaction mixture was flushed with a 1 N NaOH solution, flushed three times with distilled water and then purified from water, solvent and substrates residues using a vacuum evaporator at 60 °C. The final product was a light yellow viscous liquid and the reaction yield was equal to approx. 80%.

Syntheses using *p*-toluidine were carried out similarly and the amounts of substrates were as follows: *p*-toluidine—17.6 g, formaldehyde solution—24.6 mL, cresol—17.8 g. The resulting products had the form of light yellow crystals (*m*C-*p*T and *p*C-*p*T) or a light yellow viscous liquid (*o*C-*p*T). The reaction yield was equal to approx. 75–80%.

Non-crosslinked monomers of cresol-based benzoxazines are obtained as the products of syntheses. Benzoxazines are thermoset polymers, which means that monomers of benzoxazines are polymerized by heating until they are fully cured without the addition of any crosslinking agents. The polymerization reaction proceeds at an elevated temperature (above 200 °C) according to the oxazine ring-opening mechanism described in detail in the literature [20]. The simplified scheme of ring-opening polymerization using the *pC-A* benzoxazine example is presented in Figure 2.

### 2.2. Characteristics of Products

#### 2.2.1. Fourier Transform Infrared Spectroscopy

The presence of the expected functional groups was confirmed using Fourier transform infrared spectroscopy (FTIR), using a Vertex 70 spectrophotometer (Bruker Optik GmbH, Ettlingen, Germany). The products of syntheses were analyzed as a thin film between NaCl tablets. In the second analysis after heat treatment, the materials were analyzed in the form of tablets, formed by placing a mixture of anhydrous KBr (ca. 0.25 g) and 2 mg of the tested substance in a steel ring and pressing under the pressure of 10 MPa. The spectra were recorded at a resolution of 0.5 cm^−1^ in the spectral range 400–4000 cm^−1^.

#### 2.2.2. Nuclear Magnetic Resonance

1D and 2D ^1^H and ^13^C NMR spectra were recorded using an Avance II 400 MHz UltraShield Plus spectrometer (Bruker BioSpin GmbH, Rheinstetten, Germany), equipped with 5 mm broad-band multinuclear probe in CDCl_3_ (99.8%, Merck KGaA, Darmstadt, Germany). Chemical shifts (δ) for ^1^H and ^13^C were reported in ppm relative to the tetramethylsilane (TMS) peak.

#### 2.2.3. Differential Scanning Calorimetry

To determine the onset and peak temperature of polymerization, the DSC technique was selected. The measurements were performed using the DSC 204 F1 Phoenix differential calorimeter (Netzsch GmbH, Selb, Germany). All samples were analyzed in the temperature range of −60–300 °C, at a heating/cooling rate of 10 °C/min under nitrogen atmosphere. After the first heating, the samples were cooled and then reheated to ensure that the polymerization was complete and no additional heat effects were observed.

#### 2.2.4. Thermogravimetric Analysis

Thermogravimetric analysis was determined with use of the TG 209 F1 Libra (Netzsch GmbH, Selb, Germany) at a heating rate of 10 °C/min in the range of 20–900°C under nitrogen atmosphere. The samples during the measurement were placed in a ceramic alumina (Al_2_O_3_) crucible. For each benzoxazine sample, the TGA and DTG curves were recorded. The characteristic temperatures were determined for specified mass loss levels—5, 10 and 50%—along with the residual mass after the analysis. To investigate the behavior of the benzoxazine matrix and potential mass loss during abrasive composite curing in the furnace, samples of benzoxazine monomers were selected for TGA analysis.

#### 2.2.5. Headspace Coupled with Gas Chromatography Analysis

The total emission of volatile organic compounds (VOCs) from the synthesized benzoxazines during their crosslinking was tested by static headspace coupled with the gas chromatography technique (HS-GC). The measurements were carried out using a Clarus 580 gas chromatograph (PerkinElmer, Waltham, MA, USA) equipped with a flame ionization detector (FID) and coupled with the TurboMatrix HS 40 automated headspace sampler (PerkinElmer, Waltham, MA, USA). The samples of the synthesized benzoxazines were placed in 20 mL vials and sealed. The vials were then placed in an automatic sampler used to automatically introduce them to the thermostatic chamber in which they were heated under static conditions (without gas flow in the vial) for 5 min at 200 °C (180 °C for the reference sample). Subsequently, a specific volume of the gas sample from the headspace was collected and transferred to the chromatographic column via a heated transfer line. Chromatographic analysis was carried out at 210 °C. For the quantitative analysis of VOC released from the tested resins, the procedure of full evaporation from the headspace was used. In order to evaluate the reproducibility of the method, the measurements for each resin were repeated three times by preparing a new sample each time, placing and sealing it tightly in the headspace vial. The analytical procedure of the internal standard was used to determine the amount of the total VOC emitted from the investigated resins. The standard samples consisted of a known amount of the substrates (*p*-toluidine, aniline, *o*-cresol, *m*-cresol, *p*-cresol) and the solute placed in the headspace vials separately. For comparison, the standard resin, resole, used for abrasive articles production, was also tested.

### 2.3. Composite Preparation for Abrasion Tests

The composite samples for abrasion tests were prepared by mixing the benzoxazine resin and abrasive grains at a ratio of 1:4 by weight, which are the standard proportions used in the abrasive industry. The components were mixed using a mechanical mixer at a rate of 200 rpm for approx. 3 min at room temperature. White-fused alumina with a 120 mesh granulation was used as an abrasive grain. The mixture prepared this way was placed in the polytetrafluoroethylene (PTFE) mold and formed into round discs with a diameter of 23 ± 1 mm and a height of 5 ± 1 mm. The samples were then hardened according to the following temperature program: heating from 50 °C up to 200 °C at a heating rate of 0.2 °C/min, then heating at 200 °C for 10 h. This temperature program is a modified version of the program used in the abrasive industry for curing grinding wheels. In this case, the long duration of the curing process is caused by the construction of the mold (CNC machined PTFE plates placed between two thick metal plates and screwed together), which caused difficulties in terms of heat transfer. The relatively long curing and cooling time also protects the composites from defects such as cracks or deformations.

### 2.4. Abrasion Tests

Round samples with a diameter of 23 ± 1 mm and a height of 5 ± 1 mm were used for testing the abrasion. During the test, samples were mounted in the device perpendicular to the ground so that they simulate the work of real abrasive discs used for cutting metal bars. Before testing, the initial weight and diameter of the composite samples as well as the weight of the processed material were measured for each test set.

Steel plates were used as the material for testing the abrasion. Each time, the measurement was carried out at a constant rotational speed of the disc equal to 2650 rpm, with a constant pressure of the sample on the material of 5 kg (50 N) during 10 s. After the measurement, the loss of mass and diameter of the sample was measured, and weight loss of the steel plate was determined.

## 3. Results and Discussion

### 3.1. Fourier Transform Infrared Spectroscopy

Fourier transform infrared spectroscopy allowed us to confirm the presence of expected functional groups. The spectra obtained for all samples were characterized by a similar pattern. The exemplary spectrum of *p*C-A benzoxazine is presented in Figure 3; FTIR spectra of other synthesized benzoxazines are contained in the Appendix A (see Appendix A). Characteristic wavenumbers and corresponding vibration types of all tested samples are given in Table 2.

In all spectra, a series of characteristic signals can be observed. Absorption bands assigned to C–H_ar_ stretching vibrations appeared at 3014–3070 cm^−1^. C–H_x_ stretching vibrations from aliphatic (methyl and methylene) groups are present at 2914–2893 cm^−1^. Absorption of C=C_ar_ appeared at 1585–1612 cm^−1^ and 1467–1512 cm^−1^. Signals from the benzoxazine structure were observed at 1217–1253 cm^−1^ (asymmetric stretching C–O–C) and at 931–950 cm^−1^ (oxazine ring –O–C–N–) [33]. After curing in a furnace at 200 °C for 5 min, the signal at ~950 cm^−1^ disappeared in all samples, which additionally confirms the occurrence of ring-opening polymerization.

### 3.2. Nuclear Magnetic Resonance

Structure verification of cresol-based benzoxazines was performed in series of ^1^H and ^13^C NMR measurements. Unambiguous assignment of ^1^H and ^13^C resonances accomplished by a thorough analysis of 2D spectra (^1^H-^1^H COSY, ^1^H-^13^C gHSQC and gHMBC) and NMR data and spectra is provided in Table 3 and Table 4.

The ^1^H-NMR and ^13^C-NMR spectra of *p*C-A are shown in Figure 4 and Figure 5. Two-dimensional spectra of *p*C-A, which are ^1^H-^1^H COSY, ^1^H-^13^C gHSQC and gHMBC, are presented in Figure 6, Figure 7 and Figure 8, respectively. The full set of NMR spectra recorded for the rest of examined benzoxazines are provided in the Appendix A (see Appendix A).

Thorough analysis of ^1^H and ^13^C chemical shifts of cresol-based benzoxazines revealed significant differences within the phenyl ring, depending on the ortho-, meta- or para- position of the methyl group in the cresol part or its absence in the para- position of the aniline part. In the case of *p*C-A or *m*C-A, the splitting pattern of aromatic protons of the cresol part was similar—broad singlets were observed for H5 and H8, respectively, and two separate doublets corresponding to a pair of coupled nuclei, H7–H8 or H5–H6, respectively. The same splitting patterns were observed for *p*C-*p*T and *m*C-*p*T. Proton spectra of *o*C-A and *o*C-*p*T included three separate doublets corresponding to H5, H6 and H7. The ^1^H chemical shifts of the aniline or *p*-toluidine part were observed in the downfield part of aromatic region as three or two signals, respectively. The resonance frequencies for H5–H8 and H2′/2′′-H4′ were unambiguously assigned based on the long-range correlation peaks analysis using ^1^H-^13^C gHMBC spectra. Two broad singlets at 5.40 ± 0.1 or 4.60 ± 0.1 ppm correspond to two methylene bridges of the oxazine ring i.e., H2 and H4, respectively. Amongst the numerous ^1^H-^13^C long-range correlation peaks between H2/H4 and ^13^C of phenyl rings visible in ^1^H-^13^C gHMBC spectra, only one correlation between C4 (ca 50.7 ppm) and H5 (6.90 ± 0.1 ppm) was decisive for correct assignment of H4 and then C4, H2 and C2 signals of all compounds. One or two singlets at 2.30 ± 0.10 ppm were assigned to one or two methyl substituents of cresol-aniline or cresol-toluidine analogues, respectively.

In the ^13^C NMR spectra, most of the ^13^C signals were observed in aromatic regions (~154–116 ppm), and among them, the greatest (8–11 ppm) downfield changes of chemical shift concern quaternary carbon, i.e., C6-C8 in the phenyl ring substituted with methyl. Two ^13^C signals at 50.5 ± 0.3 ppm and 79.8 ± 0.3 ppm correspond to C2 and C4, respectively. The structure of the benzoxazine moiety was confirmed by a set of ^1^H-^13^C long range correlations between H4/H2 and quaternary carbons of the phenyl ring, i.e., of C1′ and C8a and also between H4 and C5 or C4a. Additionally, two ^1^H-^13^C connections were observed, i.e., C4-H2 and C2-H4, thus, the presence of a benzoxazine moiety connecting two phenyl rings was confirmed.

### 3.3. Differential Scanning Calorimetry

Thermal effects during heating of the monomers were measured using the DSC analysis. The exemplary DSC curve recorded for *p*C-A is shown in Figure 9.

An endotherm was observed at 106 °C and 64 °C, only for *m*C-*p*T and *p*C-*p*T, respectively, while the endothermic peak observed for *m*C-*p*T may be attributed to sublimation of substrates or potential evaporation of solvent residues. According to the literature [34], the lower enthalpy DSC peak observed for *p*C-*p*T suggests that the melting process occurred. It should be underlined that both of the *p*-toluidine-based benzoxazines were achieved in the form of the solids. For all samples, an exotherm occurred due to polymerization with onset temperature in the range between 200–240 °C, with the maximum peak at approx. 220–260 °C. Benzoxazines based on *o*-cresol exhibit the highest onset temperature as well as a maximum peak. Thermal properties of *m*C-A, *p*C-A and *p*C-*p*T are at a similar level with onset under 210 °C and a peak under 230 °C. The highest value of polymerization enthalpy (270–290 J/g) was recorded for *m*-cresol-based benzoxazines—*m*C-*p*T and *p*C-*p*T. No additional energy effects were observed in any of the samples during cooling and reheating. Summarized data collected during DSC measurement are presented in Table 5. A similar tendency in the exothermic enthalpy values was also observed for allylamine-based benzoxazines as reported by Takeichi et al. [35]. The authors also observed the lowest enthalpy for compositions containing *o*-cresol and the highest for *m*-cresol.

### 3.4. Thermogravimetric Analysis

Thermal stability is one of the key factors which qualify a given material for potential use as a component for manufacturing of grinding tools. Abrasive composites heat up to very high temperatures as a result of friction during operation; therefore, an appropriately high thermal stability of binder is particularly important [36].

The TGA curves recorded for all samples indicate mass loss of approx. 5% in the relatively broad range between 145 °C for *p*C-*p*T and up to 190 °C for *m*C-*p*T. The temperature range at which the mass loss is equal to 10% is slightly narrower and starts at ca. 180 °C (*p*C-*p*T) with the finish at ~209 °C for *m*C-*p*T. The remaining samples lost 10% of mass in the middle of the aforementioned temperature range. More significant differences between the samples were observed as the temperature increased. Benzoxazines based on *m*-cresol and *p*-cresol lost 50% of the sample mass at the temperature over 340 °C, while for *o*C-A and *o*C-*p*T, half of the mass already disappeared at approx. 240 °C. The benzoxazines based on *m*-cresol—*m*C-A and *m*C-*p*T—exhibited the highest residual mass (approx. 20%) after the analysis. TGA and DTG curves recorded for *p*C-A benzoxazine are presented in Figure 10. All characteristic data collected for the analyzed benzoxazines are summarized in Table 6.

Heat from friction during the grinding process can raise the temperature of the surface of tool up to 1000 °C [36]. In order to limit the amount of heat generated during processing, additional coolants are often used. The use of a coolant can lower the temperature of the process to 100–150 °C [37]. All the samples exhibited only 5% mass loss at 145 °C and higher, which does not exclude the use of benzoxazine based on cresol isomers as a potential binder in abrasive composites. According to Takeichi [35], it can be supposed that the significantly higher thermal degradation temperature values observed at different stages of benzoxazines decomposition for *m*C-A and *m*C-*p*T may be a result of higher crosslink density of those resins. This phenomenon may be related to the observed improved polymerization enthalpy of *m*-cresol-based bezoxazine series.

The mass percent of the emitted VOC from the studied benzoxazines at 200 °C is equal to approx. 4%, and there are no significant differences between resins obtained with *o*-cresol and *p*-cresol (see Table 7). However, the resins obtained with *m*-cresol exhibit a fundamentally different emission and, additionally, the emission in case of resins based on *m*-cresol strongly depends on the amine used: for *m*C-A, the emission at 200 °C is the highest among all benzoxazines, whereas the emission observed for *m*C-*p*T is the lowest. The lowest emission of VOC from *m*C-*p*T is in agreement with the highest thermal stability of *m*C-*p*T observed during the TGA analysis. The estimated mass percent of VOC emitted from the studied benzoxazines can suggest that the mass loss results from the emission of VOC and only a max. 1% may be the water (5% of mass loss observed at about the temperature range of 145 °C to 190 °C). It should be pointed out that the emission of VOC from the presented benzoxazines is much lower than for resole—a standard used for abrasive articles production.

### 3.5. Abrasion Tests

Data collected from abrasion tests are presented in Table 8. Samples with *o*C-A, *m*C-A and *p*C-A benzoxazine binders exhibited higher sample abrasion (*V*) than the reference sample with a phenolic resin binder. The highest sample abrasion (composite weight loss) was observed for the *o*C-A composite. The highest steel abrasion (material removal rate—MRR) among benzoxazine-based composites occurred for *m*C-A. The abrasion process is accompanied by high thermal effects and best abrasion properties of *m*C-A-based composite sample correlate with the highest thermal stability during the TGA experiment. All tested samples were characterized by a higher sample abrasion and lower MRR than the reference sample. This cannot be considered a disadvantage, because it is common practice to use softer grinding tools to machine hard materials and high-performance metal alloys [38,39,40]. The use of cresol-based benzoxazine as a binder can modify and adjust the properties of the abrasive tool to the specific properties of the machined material. It should be noted that all composites with *p*-toluidine-based benzoxazines could not be tested because they broke when trying to mount them due to high brittleness.

## 4. Conclusions

The results presented in this study demonstrate that cresol-based benzoxazines can be synthesized according to the Mannich reaction mechanism with good efficiency. Extensive structural analysis using FTIR and NMR techniques confirmed that the resulting compounds possessed the expected chemical structure and functional groups. The signals originating from the -O-C-N- structure at ~950 cm^−1^ wavenumber observed in the FTIR spectra, as well as chemical shifts of the protons present in the NMR spectra at ~4.6 ppm and ~5.4 ppm and carbons at ~50 ppm and ~80 ppm, originating from the oxazine ring methylene bridges, were the most helpful in the identification of the synthesized compounds as benzoxazines. Moreover, the effect of ring-opening polymerization was also observed on the basis of FTIR spectra in which the signal from oxazine ring disappears after curing in 200 °C. DSC analysis revealed that the thermal properties of benzoxazines based on different isomers of cresol significantly differ from each other. The measured polymerization enthalpy was in the range from 143.2 J/g for *o*C-*p*T to 287.8 J/g for *m*C-A. The highest onset temperature (~240 °C) and polymerization peak temperature (256 °C) were recorded for both of the *o*-cresol-based benzoxazines (*o*C-A and *o*C-*p*T). The change of the amine also had an impact on the thermal properties and stability, but it was not that notable. The highest polymerization enthalpy and thermal stability were observed for benzoxazines based on *m*-cresol; the residual mass after the TGA analysis for *m*C-A and *m*C-*p*T was equal to 21.2% and 18.8%, respectively. All of the benzoxazines exhibited notably lower emission of harmful VOCs during the HS-GC experiment, which is undoubtedly beneficial from the economic and ecological point of view. Lower emission corresponds to lower environmental fees for the manufacturers. The use of benzoxazines instead of the phenolic binder reduced the VOC emission by at least 46% (in the case of *m*C-A) to even 77% for *m*C-*p*T. Model abrasive composites with the cresol-based benzoxazine matrix exhibited lower MRR than the reference sample. Among the tested benzoxazine-based composites, the highest efficiency of grinding was exhibited by the *m*C-A sample, which correlates with the highest thermal stability of this resin and indicates that it is one of the most important properties of the matrix in such demanding applications. There is a wide variety of potential substrates and the possibility of modification gives almost unlimited design capabilities. All of the above conclusions allow to assume that benzoxazine resins will be gaining increasing interest from the abrasive industry as a new-generation polymer matrix for modern abrasive composites.

## Figures and Tables

**Figure 1 materials-13-02995-f001:**
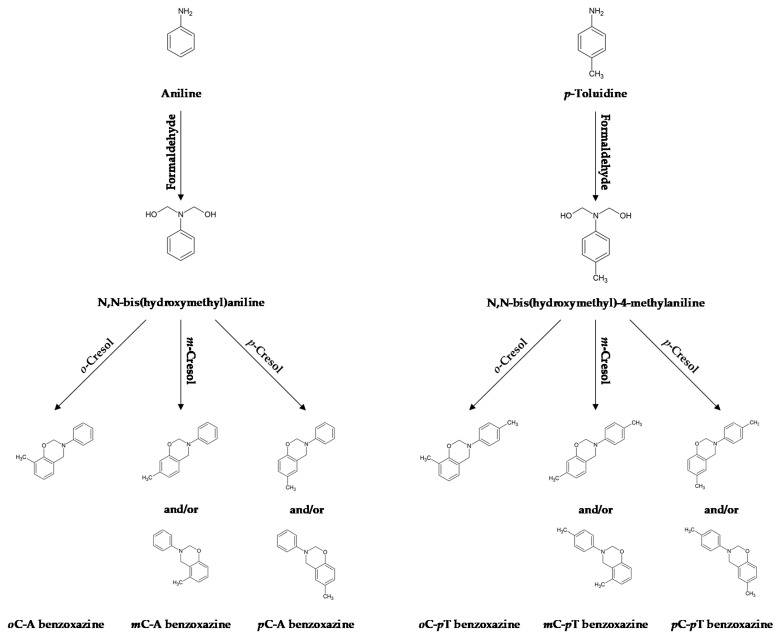
Schematic visualization of performed syntheses.

**Figure 2 materials-13-02995-f002:**
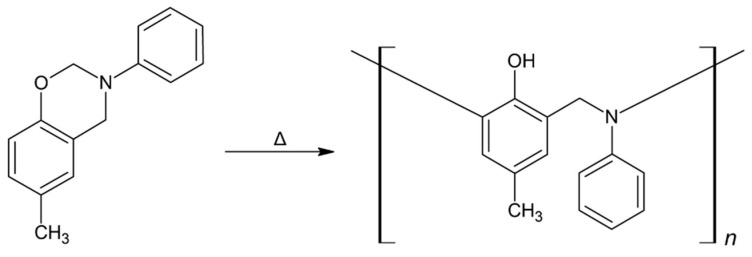
Ring-opening polymerization of *p*C-A benzoxazine.

**Figure 3 materials-13-02995-f003:**
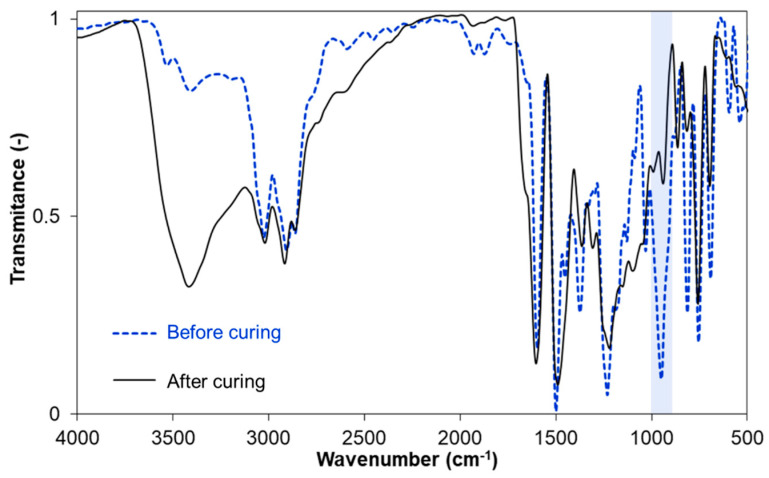
FTIR spectrum of *p*C-A benzoxazine.

**Figure 4 materials-13-02995-f004:**
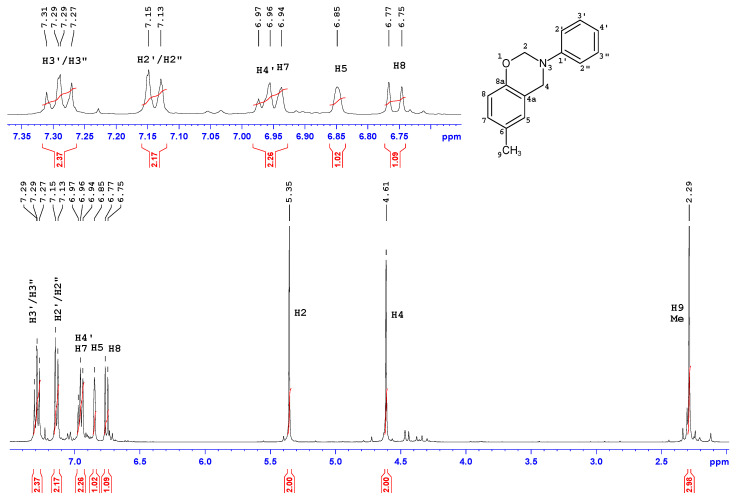
^1^H-NMR spectrum of *p*C-A benzoxazine.

**Figure 5 materials-13-02995-f005:**
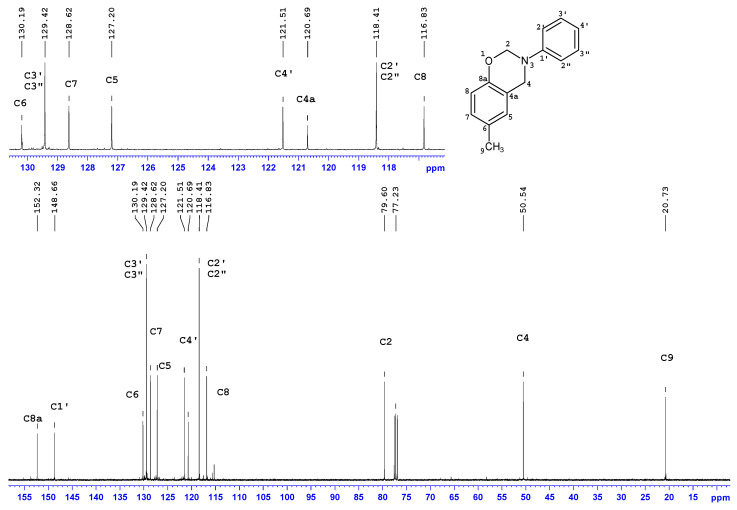
^13^C-NMR spectrum of *p*C-A benzoxazine.

**Figure 6 materials-13-02995-f006:**
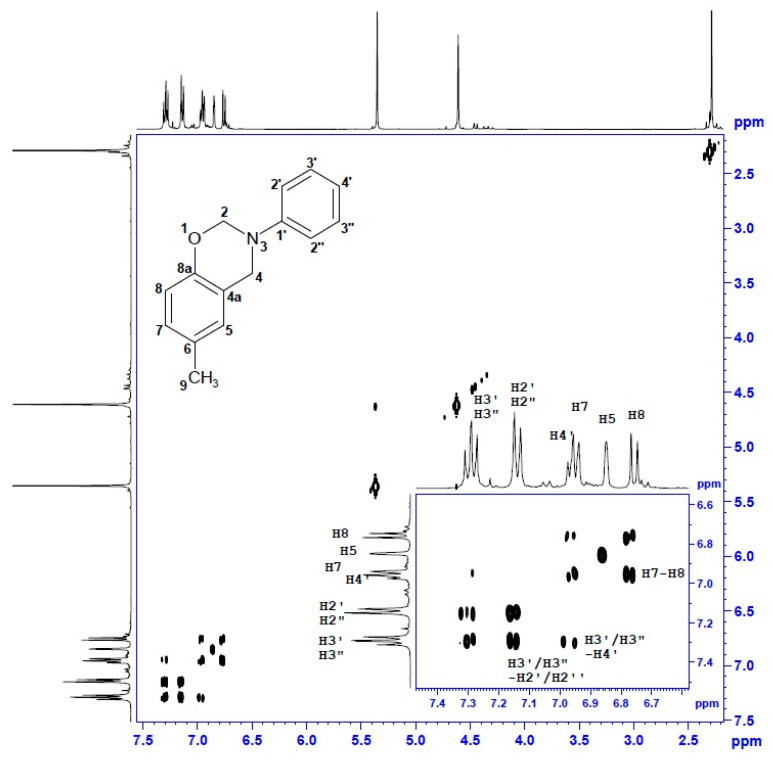
^1^H-^1^H COSY spectrum of *p*C-A benzoxazine.

**Figure 7 materials-13-02995-f007:**
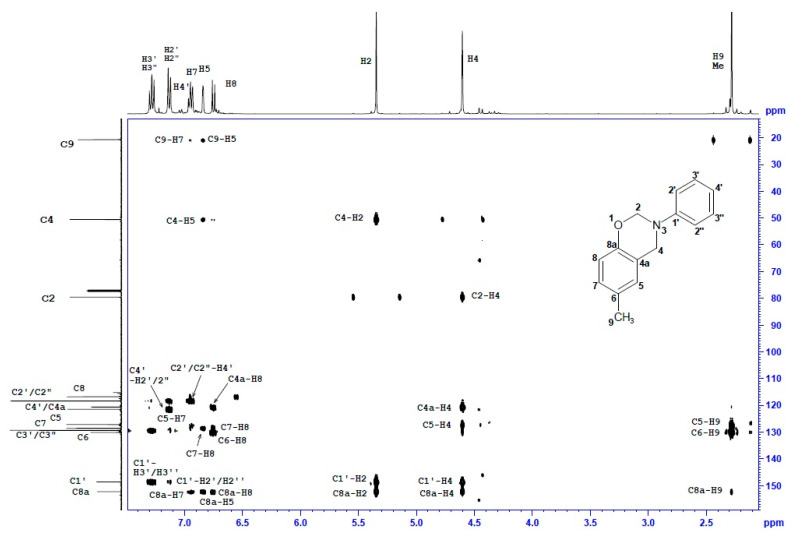
^1^H-^13^C gHSQC spectrum of *p*C-A benzoxazine.

**Figure 8 materials-13-02995-f008:**
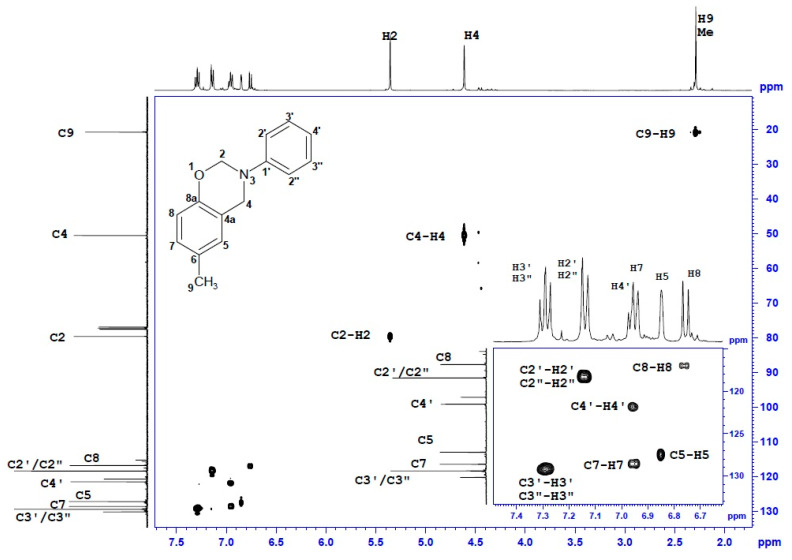
^1^H-^13^C gHMBC spectrum of *p*C-A benzoxazine.

**Figure 9 materials-13-02995-f009:**
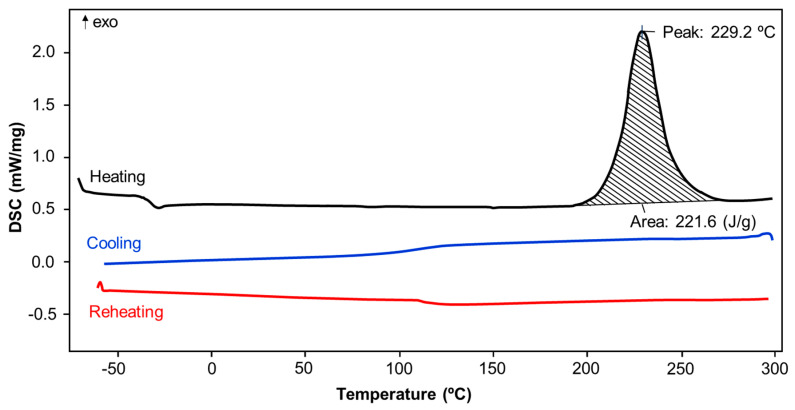
DSC curve recorded for *p*C-A benzoxazine.

**Figure 10 materials-13-02995-f010:**
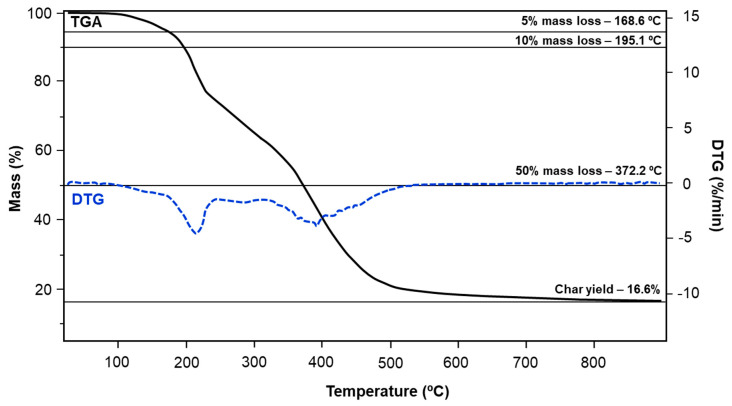
TGA and DTG curves recorded for *p*C-A polybenzoxazine.

**Table 1 materials-13-02995-t001:** Substrates used for syntheses.

Benzoxazine	Substrates		
	Cresol	Amine	Aldehyde
*o*C-A	*o*-Cresol	Aniline	Formaldehyde
*m*C-A	*m*-Cresol		
*p*C-A	*p*-Cresol		
*o*C-*p*T	*o*-Cresol	*p*-Toluidine	
*m*C-*p*T	*m*-Cresol		
*p*C-*p*T	*p*-Cresol		

**Table 2 materials-13-02995-t002:** Characteristic vibrations of functional groups with corresponding wavenumbers.

Wavenumber (cm^−1^)						Vibrations
*o*C-A	*m*C-A	*p*C-A	*o*C-*p*T	*m*C-*p*T	*p*C-*p*T	
690	690	690	657	763	663	bending C–H_ar._
757	752	752	754		736	
943	950	945	939	946	941	–O–C–N–
1217	1247	1228	1242	1253	1220	stretching C–O–C
1480	1498	1496	1467	1512	1498	stretching C=C_ar._
1608	1589	1585	1602	1585	1612	
2893	2893	2900	2914	2904	2912	stretching C–H_alif._
3040	3062	3004	3070	3014	3014	stretching C–H_ar._

**Table 3 materials-13-02995-t003:** Chemical shifts with corresponding hydrogen atoms present in the analyzed materials.

Chemical Shift (ppm)						Hydrogen Atoms Corresponding to Individual Chemical Shifts
*o*C-A	*m*C-A	*p*C-A	*o*C-*p*T	*m*C-*p*T	*p*C-*p*T	
2.33	2.35	2.32	2.26; 2.34	2.30; (x2)	2.29; 2.31	isolated -CH_3_ groups
4.69	4.66	4.64	4.64	4.59	4.59	methylene bridge from oxazine ring Ar-CH_2_-N-
5.46	5.41	5.38	5.41	5.35	5.34	methylene bridge from oxazine ring -O-CH_2_-N-
6.87–7.37	6.74–7.36	6.78–7.26	6.85–7.15	6.60–7.11	6.74–7.12	signals from protons of aromatic rings

**Table 4 materials-13-02995-t004:** Chemical shifts with corresponding carbon atoms present in the analyzed materials.

Chemical Shift (ppm)						Carbon Atoms Corresponding to Individual Chemical Shifts
*o*C-A	*m*C-A	*p*C-A	*o*C-*p*T	*m*C-*p*T	*p*C-*p*T	
15.7	21.2	20.6	15.7; 20.6	20.5; 21.1	20.6; 20.7	(s) isolated -CH_3_ groups from cresol and *p*-toluidine molecule
50.4	50.2	50.5	50.7	50.4	50.7	(s) methylene bridge from oxazine ring Ar-CH_2_-N-
79.4	79.4	79.5	80.0	79.9	80.1	(s) methylene bridge from oxazine ring -O-CH_2_-N-
118.2–152.6	117.3–154.2	116.7–152.2	118.6–152.7	117.2–154.1	116.7–152.2	(m) signals from protons of aromatic rings

**Table 5 materials-13-02995-t005:** Polymerization heat effects observed for the analyzed benzoxazines.

Analyzed Benzoxazine						Characteristic Heat Effect
*o*C-A	*m*C-A	*p*C-A	*o*C-*p*T	*m*C-*p*T	*p*C-*p*T	
240	209	210	235	238	202	Onset temperature (°C)
256	222	229	256	247	224	Maximum peak temperature (°C)
155.1	287.8	221.6	143.2	272.4	216.7	Polymerization enthalpy (J/g)

**Table 6 materials-13-02995-t006:** Mass loss stages with corresponding temperatures observed for the analyzed benzoxazines.

TGA Characteristic Temperatures of Benzoxazines						Specific Stages of TGA Analysis
*o*C-A	*m*C-A	*p*C-A	*o*C-*p*T	*m*C-*p*T	*p*C-*p*T	
166.1	156.0	168.6	147.8	189.4	145.4	5% mass loss (°C)
192.3	194.5	195.1	187.7	208.8	180.7	10% mass loss (°C)
243.7	378.7	372.2	240.1	380.4	342.5	50% mass loss (°C)
10.5	21.2	16.6	8.4	18.8	10.9	Char yield (900 °C) (%)

**Table 7 materials-13-02995-t007:** Mass percent of VOC emitted from benzoxazines at 200 °C estimated by HS-GC analysis.

	oC-A	mC-A	pC-A	oC-pT	mC-pT	pC-pT	Resole (180 °C)
Total VOC (mass %)	4.0 ± 0.3	6.9 ± 0.4	4.1 ± 0.4	3.9 ± 0.2	2.9 ± 0.5	3.9 ± 0.3	12.7 ± 0.6

**Table 8 materials-13-02995-t008:** Abrasive parameters of the tested model composites.

Sample	*V* (g/min)		*V* (mm/min)		*MRR* (g/min)	
	Value	±	Value	±	Value	±
*o*C-A	6.14	1.77	15.42	4.24	0.11	0.03
*m*C-A	2.26	0.73	5.72	3.38	0.17	0.02
*p*C-A	5.06	0.83	13.56	2.38	0.14	0.01
*o*C-*p*T	unable to test because of too brittle sample					
*m*C-*p*T	unable to test because of too brittle sample					
*p*C-*p*T	unable to test because of too brittle sample					
Reference sample	0.71	0.02	4.86	0.77	0.22	0.04

*V_p_*—sample abrasion; *V_s_*—steel abrasion.

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
