# Peer review of "Synthesis and Characterization of Low-Cost Cresol-Based Benzoxazine Resins as Potential Binders in Abrasive Composites"

_materials, 2020, doi:10.3390/ma13132995_

Round 1

Reviewer 1 Report

In this study the authors report the synthesis and characterization of cresol-based benzoxazines resins for potential application as abrasive tools. Although the topic is relevant and interesting in the field of thermoset materials for industrial applications major revisions are required before I can recommend publication, specifically to improve the scientific quality and the clarity of the presentation of the results and to better elucidate the impact of the proposed materials.

  1. In the abstract and conclusions it would be useful to add the most relevant experimental results that best describe the materials’ properties in a quantitative way in order to provide the reader with the main highlights and impact of the study.

  1. In the introduction I would suggest the authors to include more detailed quantitative information about the matrix requirements for applications as abrasive tools (such as glass transition temperature, ideal interfacial adhesion values, mechanical properties, flammability index,etc) and to correlate such properties with those of benzoxazines reported as examples in order to better highlight the potential of this class of materials for the proposed application. Moreover, the authors should better emphasize that benzoxazines have not been documented for their use as abrasive tools (in the current form, it is mentioned at the end of the introduction with no connection to the previous paragraphs).

  1. The reaction scheme is confusing especially for readers that do not have a chemistry background. I would suggest to add the name above each molecule and make two distinct reaction pathways for aniline and p-toluidine. Another suggestion would be to add the reaction yield for each final product.

  1. In line 167-168, the authors report that the composite samples for abrasion tests were prepared by mixing the benzoxazine resin and abrasive grains, at a ratio of 1:4 by weight. It seems like the abrasive grains constitute the majority of the material (4 parts against 1) although the benzoxazines should be the matix of the composite.

  1. The authors should specify the cooling rate in the DSC experimental description.

  1. The processing of benzoxazines to obtain the final cured resins should be elucidated. Are the synthesized benzoxazines polymerized by heating and then cured by further heating or is it all in one step? Do curing, hardening and cross-linking refer to the same process?

  1. The polymerization of benzoxazines should be described in the results and discussion (possibly adding a reaction scheme). In particular, the time and temperature are not well elucidated. In line 196, the authors mention that polymerization is performed 200°C for an unknown time. However, in table 5, the onset temperature for polymerization would appear to be around 200-210°C  with a peak at 220-260°C.  In line 172, the authors report that the samples comprising benzoxazines and abrasive grains were hardened according to the following temperature program: heating from 50 °C up to 200 °C heating rate 0.2 °C /min, then heating at 200 °C for 10 h. The authors perform the cross-linking of benzoxazines at 180°C for 5 minutes to perform the VOC’s analysis. All the characterizations should be performed using the same conditions. The authors should explain why they choose such conditions (in general the temperatures and times seem to greatly differ throughout the manuscript). The authors should indicate and demonstrate the optimal temperature and time to perform the polymerization/curing for each type of the synthesized benzoxazines and for the relative composites (a table could be useful to summarize the different conditions).

  1. To my understanding, the TGA is performed on the materials after curing as this would be relevant for the assessment of the thermal stability for abrasive technology applications. But this should be clarified by the authors in the manuscript as it not explicitly stated.

  1. From the abrasion test it seems that the materials undergo a greater and faster abrasion compared to the reference material and also their abrasive efficiency towards steel is lower than the reference. The authors report that this is not a disadvantage but they do not explain why. I think since this is a crucial part to demonstrate the applicability of the proposed materials, further argumentation or additional tests on different metals might be useful to demonstrate that the materials are suitable abrasive tools.

Author Response

Dear Reviewer, 

Thank You for Your insightful review of our work, which contributed to a better understanding of the scientific problems related to the subject of the publication and will help with the elimination of potential errors in the future.We would also like to express our gratitude for the revision of our manuscript and the opportunity to re-submit it, incorporating all of the Referees’ suggestions. Our comments and changes are noted below and were marked in yellow in the manuscript. 

In this study the authors report the synthesis and characterization of cresol-based benzoxazines resins for potential application as abrasive tools. Although the topic is relevant and interesting in the field of thermoset materials for industrial applications major revisions are required before I can recommend publication, specifically to improve the scientific quality and the clarity of the presentation of the results and to better elucidate the impact of the proposed materials.

Comment 1: In the abstract and conclusions it would be useful to add the most relevant experimental results that best describe the materials’ properties in a quantitative way in order to provide the reader with the main highlights and impact of the study.

Answer 1: We thank the Reviewer for this comment. Additional information specifying the values of measured parameters were added in the abstract and coclusions and marked in yellow.

Comment 2: In the introduction I would suggest the authors to include more detailed quantitative information about the matrix requirements for applications as abrasive tools (such as glass transition temperature, ideal interfacial adhesion values, mechanical properties, flammability index, etc) and to correlate such properties with those of benzoxazines reported as examples in order to better highlight the potential of this class of materials for the proposed application. Moreover, the authors should better emphasize that benzoxazines have not been documented for their use as abrasive tools (in the current form, it is mentioned at the end of the introduction with no connection to the previous paragraphs).

Answer 2: We thank Reviewer for this valuable remark. We fully agree that adding a more detailed description of the matrix requirements for abrasives would improve the understanding of the potential of proposed polymers. Unfortunately, it is highly difficult to obtain such information due to the fact that, in most cases, it is a trade secret of the manufacturer and there are not enough scientific reports concerning the relationship between matrix properties and the technical parameters of the tools. It is very difficult to point out the entry level values of the properties because tools can differ from each other in terms of shape, type and size of the abrasive grain as well as the use of additional fillers that can significantly change the properties of the tool. For example, the adhesion between polymer matrix and the abrasive grain is a property of the tool that is very difficult to determine. It is highly a complex phenomenon because if the adhesion is too weak, the grain would fall out from the tool before it loses its sharpness, which results in a shorter life cycle. In turn, if the adhesion is too strong, it will keep the grain in the mass of the tool despite the loss of sharpness, which results in lowering the grinding efficiency and raising the temperature of the operation.

We understand the Reviewer’s point of view and the fact that benzoxazines have not been documented for their use in abrasive tools is more highlighted in the introduction of the revised manuscript.

Comment 3: The reaction scheme is confusing especially for readers that do not have a chemistry background. I would suggest to add the name above each molecule and make two distinct reaction pathways for aniline and p-toluidine. Another suggestion would be to add the reaction yield for each final product.

Answer 3: We thank Reviewer for this remark, the reaction scheme was corrected according to Reviewer’s suggestion. The reaction yield of all syntheses was equal to approx. 75-80 % as it was mentioned in the manuscript, unfortunately the determination of the yield of each isomer was not the part of our scientific investigation. But we are thankful for this comment, we will include further analysis of the reaction products in the scope of our future research.

Comment 4: In line 167-168, the authors report that the composite samples for abrasion tests were prepared by mixing the benzoxazine resin and abrasive grains, at a ratio of 1:4 by weight. It seems like the abrasive grains constitute the majority of the material (4 parts against 1) although the benzoxazines should be the matix of the composite.

Answer 4: We thank Reviewer for this remark. The main purpose of the abrasive tool is the removal of the surface layer of the other, softer material. The grains that constitute the vast majority of the tool's mass are necessary to achieve high efficiency and durability of the tool. In our understanding, the matrix is the continuous phase of the composite that binds the reinforcement particles together. In our abrasive composites, benzoxazine resins that the represent polymer matrix bind the particles of the abrasive grain together. Abrasive composites are not the only example of a composite in which reinforcement particles are in the majority. One of the most common composites – asphalt concrete – consists mainly of stone, sand and gravel (almost 95 % of the total mass) and asphalt cement, which is the matrix, amounts to only approx. 5 %.

Comment 5: The authors should specify the cooling rate in the DSC experimental description.

Answer 5: We thank the Reviewer for this comment. The additional information regarding DSC cooling rate has been included in the revised manuscript version.

Comment 6: The processing of benzoxazines to obtain the final cured resins should be elucidated. Are the synthesized benzoxazines polymerized by heating and then cured by further heating or is it all in one step? Do curing, hardening and cross-linking refer to the same process?

Answer 6: We agree with the Reviewer that the processing of benzoxazines should be elucidated. After the synthesis, the non-cross-linked monomer is obtained. This single monomer of benzoxazine has an active oxazine ring, which “opens” at an elevated temperature (approx. 200 °C) and results in the cross-linking of the benzoxazine molecules that form the polymerized benzoxazine. Benzoxazines are thermoset polymers which means that the synthesized monomers of benzoxazines are polymerized by heating until they are fully cured. There is no addition of any cross-linking agents during the curing process. Curing, hardening and cross-linking refer to the same process as far as further processing of monomer is concerned.

Comment 7: The polymerization of benzoxazines should be described in the results and discussion (possibly adding a reaction scheme). In particular, the time and temperature are not well elucidated. In line 196, the authors mention that polymerization is performed 200°C for an unknown time. However, in table 5, the onset temperature for polymerization would appear to be around 200-210°C  with a peak at 220-260°C.  In line 172, the authors report that the samples comprising benzoxazines and abrasive grains were hardened according to the following temperature program: heating from 50 °C up to 200 °C heating rate 0.2 °C /min, then heating at 200 °C for 10 h. The authors perform the cross-linking of benzoxazines at 180°C for 5 minutes to perform the VOC’s analysis. All the characterizations should be performed using the same conditions. The authors should explain why they choose such conditions (in general the temperatures and times seem to greatly differ throughout the manuscript). The authors should indicate and demonstrate the optimal temperature and time to perform the polymerization/curing for each type of the synthesized benzoxazines and for the relative composites (a table could be useful to summarize the different conditions).

Answer 7: We thank reviewer for this valuable comment. We understand that such accumulation of various conditions can be confusing. The necessary corrections were introduced into the revised version of the manuscript and they are marked in yellow.

The polymerization mentioned in line 196 was performed at 200 °C due to the high concentration and small sample size (approx. 250 mg). The only reason for this was to compare the FTIR spectra of the synthesized monomers with polymerized benzoxazines. As it can be seen based on the spectra, the polymerization was successful. Relevant information regarding the duration of this process was added to the manuscript.

In line 172 there is an information regarding the temperature program we used for the curing of composite samples. This temperature program is a modified version of the temperature program used in the abrasive industry. The longer duration of the program is caused by higher sample mass of the composites in comparison with samples for FTIR and HS-GC experiments and the construction of the mold (the mold was CNC machined PTFE plates placed between two metal plates and screwed together) which caused difficulties in heat transfer. Our experience also shows that too fast curing or cooling after curing of the composites can cause defects, such as cracks or deformations.

An error has appeared in the description of VOC’s analysis. The temperature at 180 °C concerned the reference sample based on the phenol-formaldehyde resin containing 9 % of HMTA (hexamethylenetetramine). It is the standard temperature at which this type of resins and the abrasive composites with phenolic binder are cured. However, all of the benzoxazines analyzed in this experiment were conditioned at 200 °C for 5 minutes during the VOC analysis. Proper corrections were introduced into the revised version of the manuscript.

Comment 8: To my understanding, the TGA is performed on the materials after curing as this would be relevant for the assessment of the thermal stability for abrasive technology applications. But this should be clarified by the authors in the manuscript as it not explicitly stated.

Answer 8: We thank reviewer for this remark. We fully agree that it should be clearly stated which materials were subjected to TGA. The TGA was performed for the synthesized monomers of the benzoxazines. We wanted to characterize the thermal properties of the obtained monomers benzoxazines and measure the potential mass loss of the abrasive composite binder during curing. Relevant information was added in the revised version of the manuscript. However, at temperatures above 200 °C, the thermal behavior of the cross-linked polybenzoxazines can be observed.

Comment 9: From the abrasion test it seems that the materials undergo a greater and faster abrasion compared to the reference material and also their abrasive efficiency towards steel is lower than the reference. The authors report that this is not a disadvantage but they do not explain why. I think since this is a crucial part to demonstrate the applicability of the proposed materials, further argumentation or additional tests on different metals might be useful to demonstrate that the materials are suitable abrasive tools.

Answer 9: We thank the Reviewer for this comment. The abrasion results obtained for the tested materials are lower than those of currently used commercial abrasives. However, as indicated in the text, it is common practice to use softer grinding tools to machine hard materials and high performance metal alloys [38–40]. The use of cresol-based benzoxazine as a binder can modify and adjust the properties of the abrasive tool to the specific properties of the machined material. In most cases it is a part of technical experience and know-how of the manufacturer regarding the adjustment of the hardness of the tool to the specific operation and machined material. We are very thankful for the suggestion to test benzoxazine-based grinding tools on different metals, an additional abrasion tests will be the scope of our future investigation.

We look forward to hearing from you.

Yours faithfully,

Artur Jamrozik and Beata Strzemiecka

corresponding authors

Poznan University of Technology

Reviewer 2 Report

The scope of the paper is very clear, experiment well sought out, and experiment carefully done. There appears to be no technical problems. I recommend it publish on Materilas with a minor revision because the manuscript need be improved and the specific comments are as follows:

  1. Some characterization results, such as NMR and FT-IR spectra, and DSC thermograms should be provided in an additional file of Supporting Information.
  2. Figure 8

What is the temperature of the endotherm? Some more discussion should be given in this section.

  1. I suggest the authors could pay some attention on the methylol functional benzoxazine monomers (Reactive and Functional Polymers, 2018, 129, 23-28.), which may have better performance for the application as binders.

Author Response

Dear Reviewer, 

Thank You for Your insightful review of our work, which contributed to a better understanding of the scientific problems related to the subject of the publication and will help with the elimination of potential errors in the future.We would also like to express our gratitude for the revision of our manuscript and the opportunity to re-submit it, incorporating all of the Referees’ suggestions. Our comments and changes are noted below and were marked in yellow in the manuscript. 

The scope of the paper is very clear, experiment well sought out, and experiment carefully done. There appears to be no technical problems. I recommend it publish on Materials with a minor revision because the manuscript need be improved and the specific comments are as follows:

Comment 1: Some characterization results, such as NMR and FT-IR spectra, and DSC thermograms should be provided in an additional file of Supporting Information.

Answer 1: We thank the Reviewer for this remark. Additional results of the performed analyses (NMR and FTIR spectra) were compiled in the Supplementary Information file.

Comment 2: Figure 8: What is the temperature of the endotherm? Some more discussion should be given in this section.

Answer 2:  We thank the Reviewer for this comment. Figure 8 shows the thermograms obtained during the DSC experiment for pC-A beznoxazine, for which the mentioned endothermic effect was not observed. The endothermic effect at 106 °C and 64 °C occurs only for materials based on p-toluidine as well as m- and p-cresol, respectively. All samples were subjected to the DSC test in the form of solids. Therefore it can be assumed that the observed endothermic peak is the result of the melting process. The additional explanation and suitable reference have been included in a revised version of the manuscript.

Comment 3: I suggest the authors could pay some attention on the methylol functional benzoxazine monomers (Reactive and Functional Polymers, 2018, 129, 23-28.), which may have better performance for the application as binders.

Answer 3: We thank the Reviewer for this valuable remark. It is a very interesting report. We will make attempts to utilize this type of resin in our future research work. We also added the reference in the abstract of the revised version of the manuscript.

We look forward to hearing from you.

Yours faithfully,

Artur Jamrozik and Beata Strzemiecka

corresponding authors

Poznan University of Technology

Reviewer 3 Report

This manuscript investigates the synthesis and characterisation of cresol-based benzoxazine resins for their potential applications to abrasive composites. Six benzoxazine resins were synthesised using different monomers, and characterised through FTIR, NMR, DSC, TGA, and HS-GC. Finally, the resins were used to produce composite samples and tested for their abrasion performance. The overall structure of the manuscript is sound, containing very detailed descriptions of the proposed methods and evaluations. The reviewer suggests very minor corrections before publication.

Section 2.1 what is the production rates for the synthesised benzoxazine products.

The authors used different terms to represent the synthesised benzoxazines from various monomers , it would be helpful to make them clearer in Figure 1.

Author Response

Dear Reviewer, 

Thank You for Your insightful review of our work, which contributed to a better understanding of the scientific problems related to the subject of the publication and will help with the elimination of potential errors in the future.We would also like to express our gratitude for the revision of our manuscript and the opportunity to re-submit it, incorporating all of the Referees’ suggestions. Our comments and changes are noted below and were marked in yellow in the manuscript. 

This manuscript investigates the synthesis and characterisation of cresol-based benzoxazine resins for their potential applications to abrasive composites. Six benzoxazine resins were synthesised using different monomers, and characterised through FTIR, NMR, DSC, TGA, and HS-GC. Finally, the resins were used to produce composite samples and tested for their abrasion performance. The overall structure of the manuscript is sound, containing very detailed descriptions of the proposed methods and evaluations. The reviewer suggests very minor corrections before publication.

Comment 1: Section 2.1 what is the production rates for the synthesised benzoxazine products.

Answer 1: We thank Reviewer for this comment. The reaction yield of all syntheses was equal to approx. 75-80 %, as mentioned in the manuscript. Unfortunately, the determination of the yield of each isomer was not a part of our scientific investigation. But we are thankful for this comment, we will include further analysis of the reaction products in the scope of our future research.

Comment 2: The authors used different terms to represent the synthesised benzoxazines from various monomers , it would be helpful to make them clearer in Figure 1.

Answer 2: We thank Reviewer for this remark. The reaction scheme was corrected according to Reviewer suggestion.

We look forward to hearing from you.

Yours faithfully,

Artur Jamrozik and Beata Strzemiecka

corresponding authors

Poznan University of Technology

Round 2

Reviewer 1 Report

The manuscript has been significantly improved and is now suitable for publication.